# A modern pollen dataset from lake surface sediments on the central-western Tibetan Plateau

Qingfeng Ma[1], Liping Zhu[1,2], Jianting Ju[1], Junbo Wang[1], Yong Wang[3], Lei Huang[4], Torsten Haberzettl[5]

[1]State Key Laboratory of Tibetan Plateau Earth System, Environment and Resources (TPESER), Institute of Tibetan Plateau Research, Chinese Academy of Sciences, Beijing, 100101, China
[2]University of Chinese Academy of Sciences, Beijing, 100049, China
[3]Anhui Key Laboratory of Earth Surface Processes and Regional Response in Yangtze-Huaihe River Basin, School of Geography and Tourism, Anhui Normal University, Wuhu, 241002, China
[4]College of Resource Environment and Tourism, Capital Normal University, Beijing, 100048, China
[5]Physical Geography, Institute for Geography and Geology, University of Greifswald, Greifswald, 17489, Germany

*Correspondence to:* Liping Zhu (lpzhu@itpcas.ac.cn)

**Abstract.** Modern pollen datasets are essential for pollen-based quantitative paleoclimate (e.g. precipitation) reconstructions, which can aid to better understand recent climate change and its underlying forcing mechanisms. A modern pollen dataset based on surface sediments from 90 lakes in the shrub, meadow, steppe and desert regions of the central and western Tibetan Plateau (TP) was established to fill geographical gaps left by previous datasets. Ordination analyses of pollen data and climatic parameters revealed that annual precipitation is the dominant factor for modern pollen distribution on the central and western TP. A regional transfer function for annual precipitation was developed with the weighted averaging partial least squares (WA-PLS), which suggests a good inference power of the modern pollen dataset for annual precipitation. A case study in which the transfer function was effectively applied to a fossil pollen record from Lake Tangra Yumco on the central TP for paleoprecipitation reconstruction demonstrated the significance of the modern pollen dataset in less data region for paleoclimate change studies. Data from this study, including pollen data for each sample and information on the sampled sites (location, altitude and climate data), are openly available via the Zenodo portal (Ma et al., 2023; https://doi.org/10.5281/zenodo.8008474).

## 1 Introduction

Quantitative reconstructions of climate beyond the range of the instrumental record are required to understand climate change at global and regional scales (Nakagawa et al., 2002; Lu et al., 2011), and to identify the underlying forcing mechanisms of climate change. Pollen extracted from lake sediment sequences has frequently been used to quantitatively reconstruct paleoclimate at different scales from continental, regional to site-specific (Seppä et al, 2004; Lu et al., 2011). Available pollen-climate calibration sets consist of modern pollen data and the corresponding climate data. Modern pollen data from different sources (topsoils, moss pollsters, peat bogs and lake sediments) in one region have different relevant pollen source areas and taphonomy (Wilmshurst and McGlone, 2005; Zhao et al., 2009; Birks et al., 2010). Therefore, for quantitative

paleoclimate reconstructions based on fossil pollen in lake sediment records, modern pollen data for the calibration set should be extracted preferably from lake surface sediments if possible to ensure a high comparable quality (Birks et al., 2010).

The Tibetan Plateau (TP), known as the "Third Pole" of the Earth, covers a vast geographical area with an average elevation of more than 4000 m a.s.l. (Yao et al., 2013). Due to its large-scale climatic influence, sensitivity to climate change, and weak degree of anthropogenic influence, the TP has become a key region to study present and past climate change and the underlying forcing mechanisms. To date, there are several modern pollen datasets from the TP that have been used to develop the pollen-climate calibration sets (Shen et al., 2006; Herzschuh et al., 2010; Lu et al., 2011; Wang et al., 2014; Cao et al., 2021). However, these datasets are mostly from the eastern TP, short of samples from lake surface sediments evenly distributed across the central and western TP. A large number of evenly distributed lakes from the central and western TP provide an opportunity to establish a lake surface sediment pollen dataset for this region.

Here we present and analyse a modern pollen dataset from lake surface sediments on the central and western TP. Our aims are to (1) describe the modern pollen assemblages on the central and western TP; (2) determine the key climatic variable that is mainly responsible for controlling the pollen distribution on the central and western TP; (3) derive a regional pollen-climate transfer function to be able to quantitatively reconstruct the key climatic variable; (4) evaluate the predictive power of our pollen dataset for quantitative paleoclimate reconstruction through a case study. This modern pollen dataset not only supports the accurate regional paleoclimate reconstruction in the central and western TP, but also fills a geographical gap left by previous datasets to improve the reliability of quantitative climate reconstructions for the entire TP.

## 2 Study area

### 2.1 Topography and Climate

The TP is the broadest and highest elevation collisional system of the Earth (Ding et al., 2022). Under the long-term comprehensive effect of the internal and external agency, the TP manifests extremely complex topographic features. Our study region is located in the central and western TP between 28.5°-35.2°N and 79.8°-91.5°E. Several nearly west-east trending mountain ranges are distributed in the region, which are the Himalayas, the Gangdese-Nyainqentanglha Mountains, the Karakorum-Tanggula Mountains and the Kunlun Mountains from south to north. Between these mountain ranges there are several broad valleys, which are divided by the secondary tectogenesis into many sedimentary basins of different sizes. A large number of lakes lie on the bottom of the basins (ITCAS, 1988). A total of 90 lakes were sampled in the study region, which information is presented in Ma et al. (2023). Altitudes of the sampled lakes for surface sediment range from 4240 to 5220 m a.s.l. (Table 1). The average elevation of these lakes is 4740 m a.s.l. The area is influenced by the Indian summer monsoon in summer and the Westerlies in winter (Yao et al., 2013). The region spans over several climatic zones from the frigid arid climate zone in the northwest to the temperate arid climate zone in the west part, to the subfrigid semi-arid climate zone in the central and north, and the temperature semi-arid climate zone in the south (Institute of Geography, 1990). The study area is

characterized by cool-humid summers and cold-dry winters with mean annual temperatures between -13 and 5 ℃ and a mean annual precipitation of 50-500 mm from 1981 to 2010 (Figure 1a; He et al., 2020).

**Table 1.** Summary statistics of environmental parameters in the pollen dataset.

| Climate variable | Minimum | Maximum | Mean | Standard deviation |
|---|---|---|---|---|
| Altitude (m a.s.l.) | 4241 | 5217 | 4740 | 243 |
| $P_{ann}$ (mm) | 75 | 478 | 236 | 109 |
| $T_{ann}$ (℃) | -13 | 4.2 | -3.4 | 4.1 |
| $T_{July}$ (℃) | 0.4 | 13.4 | 7.8 | 3.3 |
| $T_{Jan}$ (℃) | -26.3 | -4 | -14.7 | 5.1 |

## 2.2 Vegetation

Vegetation mainly consists of shrubs, meadows, steppes and deserts (Figure 1b; ITCAS, 1988; Institute of Geography, 1990; Hou, 2001). Shrubs, generally including *Sophora*, *Ceratostigma*, *Rosa*, *Caragana*, *Rhododendron*, *Potentilla*, and *Berberis*, are mainly distributed below 4800 m a.s.l. in the southern part of the study area. Alpine meadows are mainly found at an

elevation of 4500-5300 m a.s.l. in the eastern part, at 4700-5200 m a.s.l. in the south, and 5100-5300 m a.s.l. in the central-southern part. They mainly consist mainly of *Kobresia pygmaea* communities, associated with plants of *Kobresia*, *Carex*, Poaceae, *Stipa*, Asteraceae, Ranunculaceae, Polygonaceae, and Fabaceae. Alpine steppes occupy the greatest part of the study area, which are distributed at 5100-5400 m a.s.l. in the northwestern part and below 5100 m a.s.l. in other parts. The composition of family and genus in the Tibetan steppe is dominated by Poaceae (e.g., *Stipa spp.*) and Asteraceae (e.g.

*Artemisia spp.*), together with Cyperaceae, Fabaceae, Caryophyllaceae, Brassicaceae and Amaranthaceae (formerly known as Chenopodiaceae, classified into the family Amaranthaceae). Alpine deserts are mainly distributed in the west and northwest (4200-5100 m a.s.l.). The plant communities are dominated by Amaranthaceae and Asteraceae, such as *Ceratoides latens*, *C. compacta*, *Ajania fruticulosa* and *Artemisia sp.*, together with Poaceae, Fabaceae, Brassicaceae, and *Ephedra* (ITCAS, 1988; Institute of Geography, 1990; Hou, 2001).

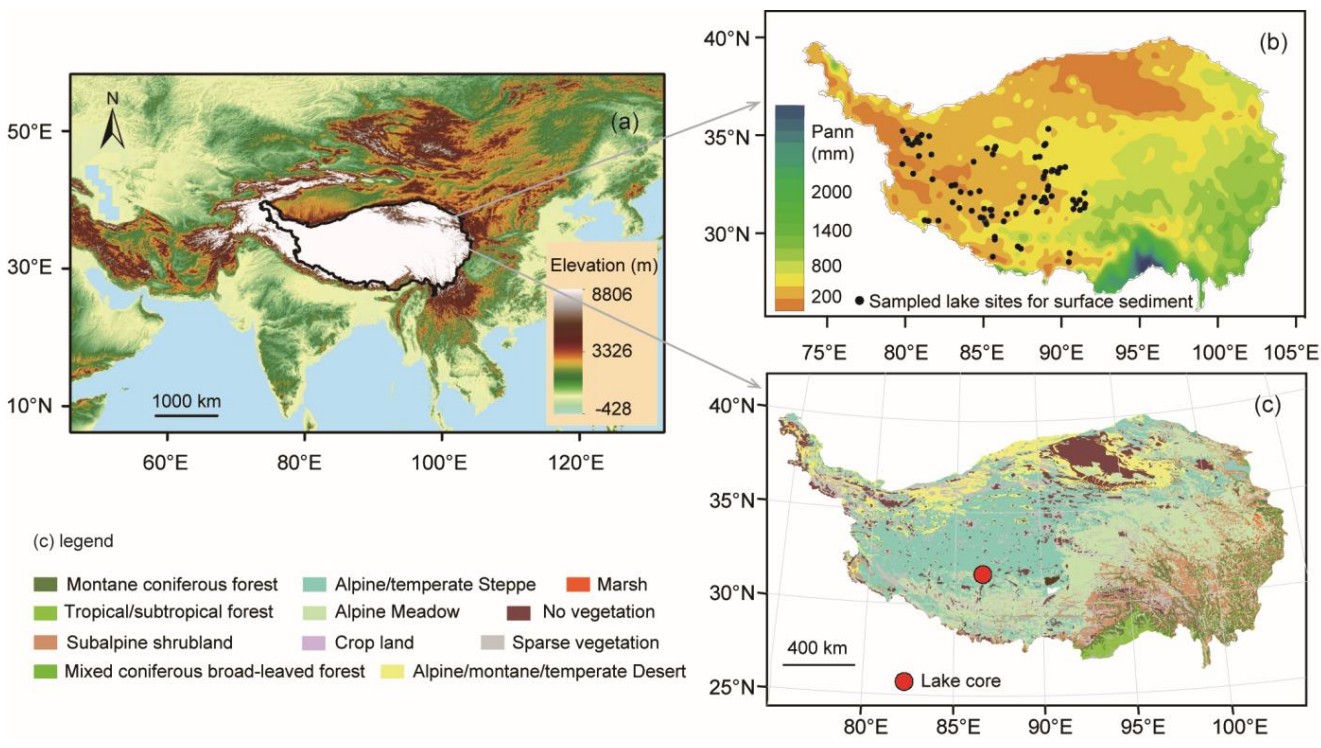

**Figure 1.** Study area and sampling sites of modern lake sediments. (a) Location of the study area; (b) spatial distribution of sampling lake sites (black dots) and annual precipitation; (c) vegetation map of the Tibetan Plateau with the location of Lake Tangra Yumco (red dot).

## 3 Materials and Methods

### 3.1 Pollen sample collection and analysis

Lake surface sediments were collected from 90 lakes in the study area, including the previously studied 31 lake surface sediment samples (Ma et al., 2017). These lakes fill the geographical gaps in the central and western TP, especially the northern Tibet, where is difficult to access due to the lack of the settlements and roads (Figure 1a). To reduce the influence of the lakeshore vegetation on the pollen spectra, surface sediments were sampled from the central part of each lake. The top 2 cm of sediment was collected as a single sample for each lake, corresponding approximately to the deposition of the last 10-20 years (Wang et al., 2020). Approximately 5 g (dry weight) of each sample was subsampled and used for pollen extraction. Samples were prepared using standard techniques (Fægri and Iversen, 1975), involving 10% HCl, 10% KOH, 40% HF, and a mixture (9:1) of acetic anhydride and sulfuric acid, followed by sieving through a 10 μm mesh. Pollen identification and counting were performed under a ZEISS microscope at 400×magnification, aided by published keys (Tang et al., 2017; Wang et al., 1995).


More than 300 pollen grains were counted for each sample. A total of 70 pollen taxa were recognized from the surface sediments of 90 lakes (Table 2).

## 3.2  Climate parameter processing

Climate data were obtained from the Chinese Meteorological Forcing Dataset (He et al., 2020). The dataset is based on the fusion of remote sensing products, reanalysis datasets and in-situ station data, with a temporal resolution of 3 hours and a
spatial resolution of 0.1 °. The gridded reanalysis/remote sensing data are GLDAS NOAH10SUBP 3H, GLDAS NOAH025 3H, Modern Era Retrospective-Analysis for Research and Applications (MERRA) MAI3CPASM 5.2.0 20, Global Energy and Water Exchanges – Surface Radiation Budget (GEWEX-SRB) REL3.0 SW 3HRLY 21, and TRMM 3B42 v7. The dataset introduces high-resolution elevation data in the interpolating process of particular variables, such as air temperature. The dataset has been positively validated for western China where observational data are sparse. Near-surface (2 m) air temperature
and precipitation rates were extracted for each lake during 1981-2010 based on their longitude and latitude. These were subsequently transformed to mean annual temperature ($T_{\mathrm{ann}}$), mean July temperature ($T_{\mathrm{July}}$), mean January temperature ($T_{\mathrm{Jan}}$) and mean annual precipitation ($P_{\mathrm{ann}}$) to estimate the influence of climate variables on modern pollen distribution. The four climate variables were selected because they are important limiting factors for pollen distribution and plant growth, are commonly presented in climate models, and are most relevant for paleoclimatic reconstructions (Lu et al., 2011; Cao et al.,
2021). Summary statistics for the four climate variables are presented in Table 1.

**Table 2.** List of the 70 pollen taxa in the modern dataset, derived from lake surface sediments (SD= Standard deviation).

| Pollen taxa | Hill's N2 | Min. | Max. | Mean | SD | Pollen taxa | Hill's N2 | Min. | Max. | Mean | SD |
|---|---|---|---|---|---|---|---|---|---|---|---|
| *Picea* | 28.80 | 0.00 | 2.90 | 0.36 | 0.52 | *Aster* | 43.68 | 0.00 | 7.15 | 1.37 | 1.41 |
| *Abies* | 12.87 | 0.00 | 1.69 | 0.11 | 0.27 | *Pedicularis* | 17.03 | 0.00 | 1.10 | 0.11 | 0.22 |
| *Pinus* | 61.67 | 0.31 | 36.60 | 10.74 | 7.28 | Caryophyllaceae | 32.91 | 0.00 | 10.80 | 1.15 | 1.51 |
| *Tsuga* | 23.16 | 0.00 | 4.75 | 0.42 | 0.73 | *Humulus* | 13.91 | 0.00 | 1.33 | 0.10 | 0.24 |
| *Cedrus* | 23.67 | 0.00 | 1.36 | 0.16 | 0.27 | Rosaceae | 44.91 | 0.00 | 17.42 | 3.52 | 3.52 |
| *Keteleeria* | 9.65 | 0.00 | 0.94 | 0.06 | 0.17 | Brassicaceae | 44.80 | 0.00 | 4.17 | 0.81 | 0.82 |
| *Betula* | 43.71 | 0.00 | 9.86 | 1.55 | 1.60 | Amaranthaceae | 34.09 | 0.00 | 64.60 | 11.59 | 14.84 |
| *Alnus* | 25.65 | 0.00 | 8.86 | 0.94 | 1.49 | Zygophyllaceae | 12.09 | 0.00 | 1.21 | 0.08 | 0.19 |
| *Ulmus* | 8.67 | 0.00 | 1.30 | 0.07 | 0.21 | Cyperaceae | 58.46 | 1.62 | 88.18 | 23.58 | 17.32 |
| *Quercus*-evergreen | 38.56 | 0.00 | 9.68 | 1.61 | 1.86 | Lamiaceae | 15.74 | 0.00 | 8.70 | 0.45 | 0.98 |
| *Corylus* | 1.00 | 0.00 | 0.76 | 0.01 | 0.08 | *Thalictrum* | 35.41 | 0.00 | 3.78 | 0.49 | 0.61 |
| *Carpinus* | 11.04 | 0.00 | 3.00 | 0.19 | 0.50 | Ranunculaceae | 35.78 | 0.00 | 5.90 | 0.97 | 1.19 |
| *Cyclobalanopsis* | 1.00 | 0.00 | 0.33 | 0.01 | 0.03 | *Saxifraga* | 14.38 | 0.00 | 1.92 | 0.20 | 0.45 |
| *Quercus*-deciduous | 12.69 | 0.00 | 3.26 | 0.22 | 0.53 | *Gentiana* | 6.69 | 0.00 | 6.41 | 0.22 | 0.79 |
| *Pterocarya* | 3.99 | 0.00 | 0.32 | 0.01 | 0.06 | *Polygonum* | 30.53 | 0.00 | 1.92 | 0.32 | 0.45 |
| *Juglans* | 16.36 | 0.00 | 0.56 | 0.06 | 0.13 | *Taraxacum* | 13.41 | 0.00 | 2.24 | 0.16 | 0.37 |
| *Rhamnus* | 3.28 | 0.00 | 0.72 | 0.02 | 0.10 | Apiaceae | 6.57 | 0.00 | 0.82 | 0.04 | 0.14 |

| | | | | | | | | | | |
|---|---|---|---|---|---|---|---|---|---|---|
| Moraceae | 1.98 | 0.00 | 0.31 | 0.01 | 0.04 | Euphorbiaceae | 12.51 | 0.00 | 6.65 | 0.39 | 0.97 |
| Celastraceae | 1.00 | 0.00 | 0.31 | 0.00 | 0.03 | Solanaceae | 2.68 | 0.00 | 0.84 | 0.02 | 0.13 |
| Theaceae | 1.00 | 0.00 | 0.20 | 0.00 | 0.02 | Boraginaceae | 1.00 | 0.00 | 0.32 | 0.00 | 0.03 |
| *Rhus* | 1.00 | 0.00 | 0.30 | 0.00 | 0.03 | *Urtica* | 3.76 | 0.00 | 3.27 | 0.08 | 0.36 |
| *Acer* | 1.00 | 0.00 | 0.32 | 0.00 | 0.03 | *Peganum* | 2.75 | 0.00 | 0.65 | 0.02 | 0.10 |
| Symplocaceae | 1.00 | 0.00 | 0.33 | 0.00 | 0.03 | Thymelaeaceae | 3.54 | 0.00 | 0.67 | 0.02 | 0.10 |
| *Salix* | 7.64 | 0.00 | 0.49 | 0.03 | 0.09 | *Stellera* | 1.00 | 0.00 | 0.29 | 0.00 | 0.03 |
| Ericaceae | 8.84 | 0.00 | 0.31 | 0.03 | 0.08 | *Rumex* | 8.04 | 0.00 | 0.69 | 0.04 | 0.11 |
| *Spiraea* | 6.29 | 0.00 | 0.90 | 0.04 | 0.14 | Scrophulariaceae | 3.71 | 0.00 | 0.54 | 0.02 | 0.08 |
| *Ephedra* | 25.89 | 0.00 | 4.86 | 0.51 | 0.80 | *Sanguisorba* | 1.99 | 0.00 | 0.35 | 0.01 | 0.05 |
| *Tamarix* | 4.87 | 0.00 | 1.20 | 0.04 | 0.16 | Rubiaceae | 7.97 | 0.00 | 3.00 | 0.13 | 0.41 |
| *Hippophae* | 29.06 | 0.00 | 7.97 | 0.68 | 0.98 | Liliaceae | 4.00 | 0.00 | 0.30 | 0.01 | 0.06 |
| *Nitraria* | 15.86 | 0.00 | 2.00 | 0.15 | 0.33 | Crassulaceae | 2.97 | 0.00 | 0.30 | 0.01 | 0.05 |
| *Elaeagnus* | 2.42 | 0.00 | 1.49 | 0.03 | 0.17 | Primulaceae | 1.00 | 0.00 | 0.32 | 0.00 | 0.03 |
| Fabaceae | 31.22 | 0.00 | 2.58 | 0.40 | 0.55 | *Erigeron* | 5.35 | 0.00 | 0.60 | 0.02 | 0.09 |
| *Artemisia* | 69.65 | 1.29 | 66.67 | 29.28 | 15.82 | Haloragaceae | 1.00 | 0.00 | 0.31 | 0.00 | 0.03 |
| Poaceae | 51.77 | 0.00 | 25.56 | 5.00 | 4.29 | Typhaceae | 1.69 | 0.00 | 0.72 | 0.01 | 0.08 |
| Asteraceae | 39.16 | 0.00 | 9.20 | 1.37 | 1.56 | *Potamogeton* | 1.95 | 0.00 | 1.29 | 0.02 | 0.17 |

## 3.3 Numerical analyses

Ordination techniques were used to identify co-varying patterns between modern pollen spectra and environmental variables. In order to stabilize the variances and to optimize the signal-to-noise ratio in the dataset, pollen percentages were square-root-transformed (Prentice, 1980). Detrended correspondence analysis (DCA) was used to determine which method (e.g. linear or non-linear method) was most suitable based on gradient length as the criterion. The DCA result showed that pollen data have a gradient length of 2.2 standard deviations. Therefore, a linear method (redundancy analysis (RDA)) was considered appropriate for our pollen data (ter Braak and Prentice, 1988; ter Braak and Verdonschot, 1995). An RDA was carried out to detect the influence of climate variables on our modern pollen dataset. Monte Carlo permutation tests (999 unrestricted permutations) were used to assess the statistical significance of the RDA models. To reduce the bias caused by the effects of high collinearity between climate variables in the ordination analysis process, we examined the variance inflation factors (VIFs) for each variable. If the VIF value of a variable was larger than 20, the variable was expected to be collinear with other variables and to capture little variance (ter Braak, 1988; ter Braak and Šmilauer, 2012). The initial RDA showed that the VIF values of the variables $T_{ann}$, $T_{Jan}$ and $T_{July}$ were greater than 20. However, after deleting $T_{ann}$ that held the highest VIF value, the remaining three climate variables had VIF values lower than 20 and could therefore be used in the final RDA to discern their influence on the modern pollen dataset. These analyses were performed using the Canoco 5 software (ter Braak and Šmilauer, 2012).

To assess the potential of the modern pollen dataset for quantitative estimates of climate parameters, the Weighted

Averaging Partial Least Squares (WA-PLS) model was run. WA-PLS is one of the most robust techniques for numerical climate inference (ter Braak and Juggins, 1993) and has been widely used in quantitative analyses of relationships between biological assemblages and environmental factors on the TP, such as on pollen (Shen et al., 2006; Herzschuh et al., 2010) and other proxy data, e.g. chironomid (Zhang et al., 2019). Error estimates of the WA-PLS model were compared with other calibration models, including weighted averaging (WA) regression with inverse deshrinking (Birks et al., 1990), modern

analogue technique (MAT) (Overpeck et al., 1985), weighted average of the k closest modern analogues (WMAT) (ter Braak, 1995), weighted averaging partial least-squares regression (WA-PLS) and tolerance-weighted WA-PLS (TWA-PLS) (Liu et al., 2020). TWA-PLS is an improved version of WA-PLS by considering the information about the climatic tolerances of taxa (Liu et al., 2020). TWA-PLS was performed using the function *TWAPLS.w* function in the package *fxTWAPLS* version 0.1.2 (Liu et al., 2020) for R 4.2.3. In addition to the traditional leave-one-out cross-validation, another method (Telford and Birks,

2011) was used to assess the statistical significance of the climate reconstruction for WA-PLS reconstruction method using the statistical package R 4.2.3 (R Core Team, 2023) with the *random*TF function in the *palaeo*Sig package 2.1-3 (Telford and Trachsel, 2023). Final model development and climate reconstructions for WA-PLS were performed in the C2 software (Juggins, 2003).

To evaluate the inference power of our pollen dataset for paleoclimatic parameters, an application of the WA-PLS model

to the Lake Tangra Yumco fossil pollen record (covering the last 17.5 cal kyr BP; Ma et al., 2019, 2020) was made to quantitatively estimate the last deglacial and Holocene paleoclimate in the south-central part of the TP. Lake Tangra Yumco (31.2 °N, 86.7 °E, 4545 m a.s.l.) is located in the semi-arid region, and its catchment is mainly occupied by alpine steppe (Figure 1b).

## 4  Pollen data and relations to climate factors

The pollen assemblages in the surface sediments of the 90 lakes, consisting of a total of 70 pollen taxa, are summarized in Table 1. The pollen assemblages (Figure 2) are dominated by herbaceous taxa, especially *Artemisia* (mean 29.3 ±15.8%, max. 66.7%), Cyperaceae (mean 23.6 ± 17.3%, max. 88.2%) and Amaranthaceae (mean 11.6 ± 14.8%, max. 64.6%). The sum percentage of these three taxa is up to 64.4%. Other common herbaceous pollen taxa include Poaceae (mean 5 ±4.3%, max. 25.6%), *Aster* (mean 1.4 ±1.4%, max. 7.2%), Asteraceae (mean 1.4 ±1.6%, max. 9.2%), Caryophyllaceae (mean 1.2 ±1.5%,

max. 10.8%), Ranunculaceae (mean 1.0 ± 1.2%, max. 5.9%), Brassicaceae (mean 0.8 ±0.8%, max. 4.2%), and *Thalictrum* (mean 0.5 ±0.6%, max. 3.8%). Tree pollen consists mainly of *Pinus* (mean 10.7 ±7.3%, max. 36.6%), *Quercus*-evergreen (*Quercus* E; mean 1.6 ±1.9%, max. 9.7%), *Betula* (mean 1.6 ±1.6%, max. 9.9%) and *Alnus* (mean 1.0 ±1.5%, max. 8.9%). Shrub pollen percentages are low, mainly consisting of Rosaceae (mean 3.5 ±3.5%, max. 17.4%), *Hippophae* (mean 0.7 ± 1.0%, max. 8.0%) and *Ephedra* (mean 0.5 ±0.8%, max. 4.9%).

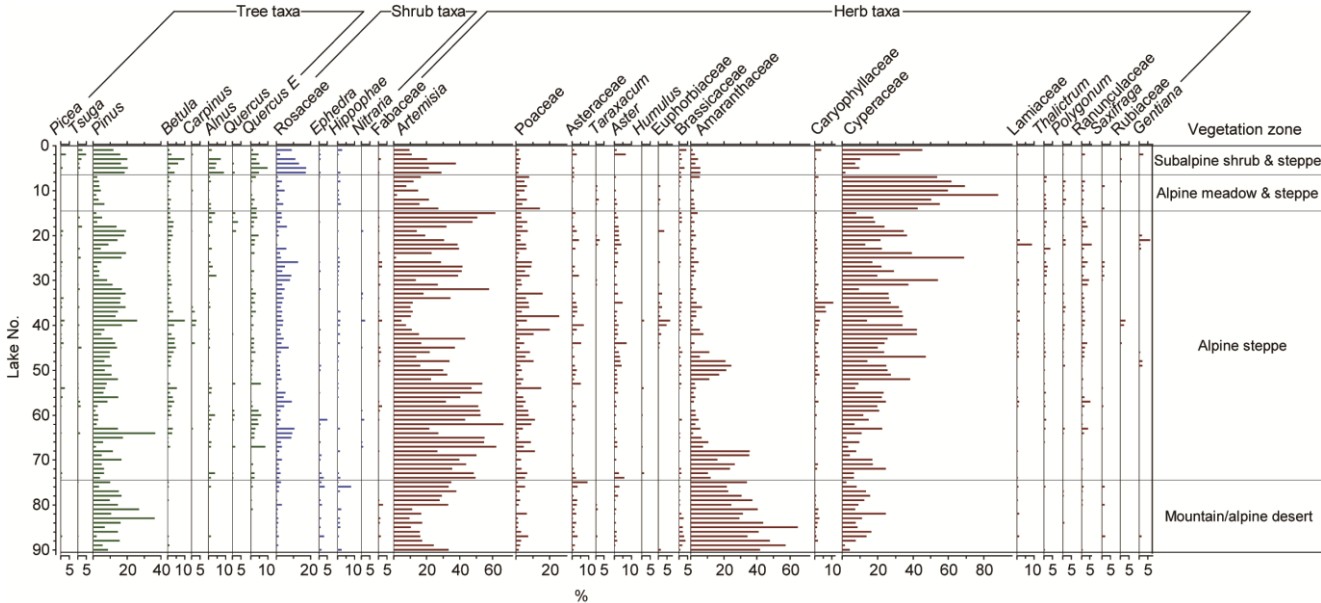


**Figure 2.** Pollen diagram for lake surface sediments from the central and western Tibetan Plateau showing the percentage values of selected pollen taxa. The samples are arranged according to their vegetation zones.

A series of RDAs based on 31 pollen taxa (1% in at least two samples) and three climatic variables ($P_{ann}$, $T_{Jan}$ and $T_{July}$) show that $P_{ann}$, as a sole variable, explains 24.2% of the variation in the pollen data, which is significantly more than the explained proportion of $T_{Jan}$ (8.6%) and $T_{July}$ (2.5%). The RDA result with all climatic variables is shown in Figure 3. RDA axis 1 and axis 2 capture 26.5% and 7.2% of the variation in the pollen data, respectively. Humid-preferring taxa including Cyperaceae, *Thalictrum*, Ranunculaceae, and *Polygonum* are located along the positive direction of $P_{ann}$, whereas arid-preferring taxa consisting of Amaranthaceae, *Ephedra*, *Artemisia*, and Asteraceae are distributed along the negative direction of $P_{ann}$. Tree pollen taxa are mainly located along the positive direction of $T_{Jan}$ and $T_{July}$.


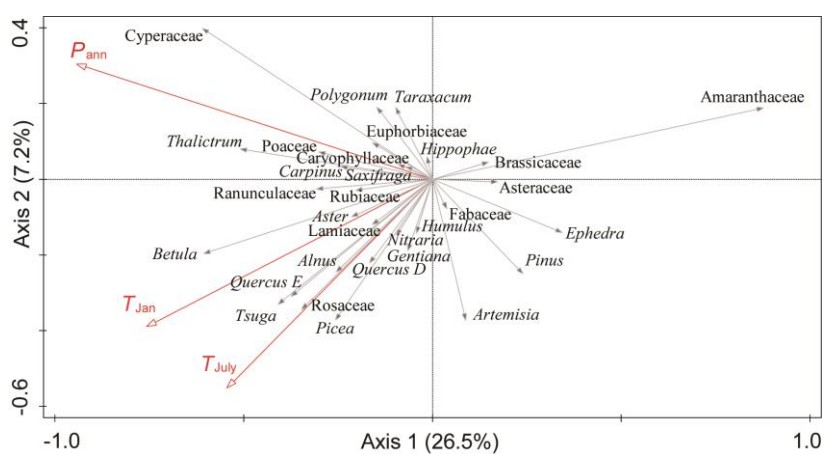

**Figure 3.** Redundancy analysis (RDA) results for the main pollen taxa in the surface lake sediments and three important climatic variables ($P_{ann}$ = mean annual precipitation; $T_{Jan}$ = January temperature; $T_{July}$ = July temperature).

## 5 Pollen-climate transfer function construction and application

RDA results indicate that $P_{ann}$ is the leading climate variable for modern pollen distribution in the central and western TP (Figure 3; Table 3). The associated Monte Carlo permutation test exhibits that the relationship between pollen taxa and $P_{ann}$ is significant ($p = 0.001$) (Table 3). Thus, the numerical relationship between modern $P_{ann}$ and pollen assemblages shows that the pollen dataset has robust ability to infer the changes in climatic variables. Performances of various calibration models are shown in Table 4. The last significant number of component for WA-PLS or TWA-PLS is considered to be the suitable

component for each method. Thus, WA-PLS component 2 ($p = 0.031$) and TWA-PLS component 2 ($p = 0.047$) are used to compare with other methods. WA-PLS component 2 and TWA-PLS component 2 show better performances than WA, MAT and WMAT, due to their lower RMSEP and higher $R^2$. Further comparison between WA-PLS and TWA-PLS reveals that WA-PLS component 2 has a lower RMSEP and a higher $R^2$. It means that TWA-PLS does not improve model performance based on our pollen dataset, compared with WA-PLS. One explanation could be that our sample size is not large enough and

sample sites are mainly distributed in middle and lower parts of the precipitation gradient with a relatively uniform distribution. Finally, we still use WA-PLS method to develop the pollen-precipitation transfer function for our dataset.

     **Table 3.** Summary statistics of redundancy analysis (RDA) based on main pollen taxa and four climate variables. VIF is the variance inflation factor (without $T_{ann}$). Variance explained (%) is the proportion of variation in the pollen assemblages explained by each variable as the sole constraining variable in RDA. $p$ value indicates the statistical significance of each

variable assessed by Monte Carlo permutation tests (999 unrestricted permutations).

| Climate variable | VIF | Variance explained (%) | $p$ |
|---|---|---|---|
| $P_{ann}$ | 3.7 | 24.2 | 0.001 |
| $T_{Jan}$ | 11.1 | 8.6 | 0.001 |
| $T_{July}$ | 6.6 | 2.5 | 0.002 |
| $T_{ann}$ | – | – | – |


**Table 4.** Error estimates for different $P_{ann}$ calibration models, assessed by leave-one-out cross-validation. $R^2$ ($R^2$_Jack) is the coefficient of determination between predicted and observed $P_{ann}$. RMSEP is the root mean square error of prediction. $p$ value for each component assesses whether using the current component is significantly different from using one component less.

| Model | Method | $R^2$ | RMSEP (mm) | $p$ |
|---|---|---|---|---|
| $P_{ann}$ (mm) | WA_Inv | 0.636 | 65.753 | – |
| | MAT | 0.629 | 67.024 | – |
| | WMAT | 0.637 | 66.28 | – |
| | WA-PLS component 1 | 0.633 | 66.609 | 0.001 |
| | WA-PLS component 2 | 0.684 | 61.381 | 0.031 |
| | WA-PLS component 3 | 0.675 | 62.502 | 0.79 |
| | TWA-PLS component 1 | 0.616 | 68.072 | 0.001 |
| | TWA-PLS component 2 | 0.675 | 62.262 | 0.047 |
| | TWA-PLS component 3 | 0.676 | 62.367 | 0.528 |

The final WA-PLS component 2 model shows a good performance ($R^2 = 0.684$, RMSEP = 61.381 mm). The results of the WA-PLS component 2 model for $P_{ann}$ with a plot of predicted versus observed values are shown in Figure 4. The results indicate that WA-PLS component 2 performs well for $P_{ann}$ inference. However, the plot of the residuals versus $P_{ann}$ values shows an underestimate for humid sites ($P_{ann} > 400$ mm). Nevertheless, quantitative reconstructions covering the arid and semi-arid region of the TP ($P_{ann} < 400$ mm) should be credible due to the low bias.


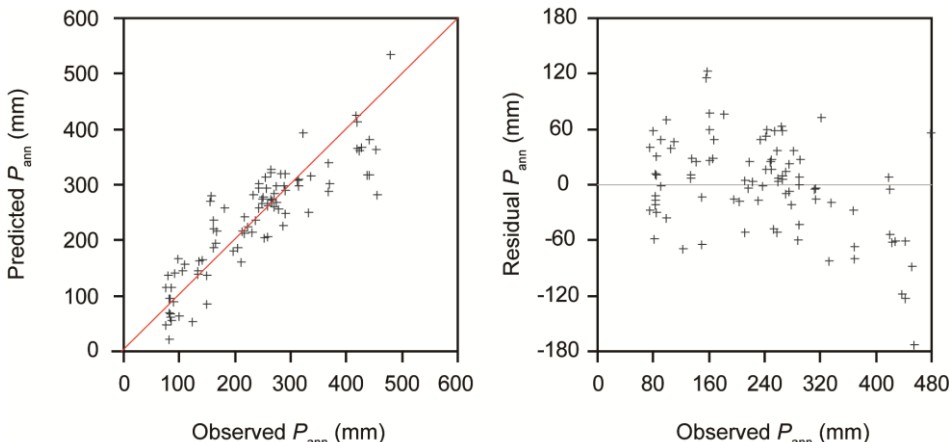

**Figure 4.** Scatter plots of WA-PLS inference model for observed annual precipitation ($P_{ann}$) vs. predicted $P_{ann}$ values (left) and residuals vs. $P_{ann}$ values (right).

Our pollen-climate transfer function can be considered as a regional calibration model for the central and western TP. So far, there is no special calibration model for this region. Compared with the large-scale calibration models covering our study region and other regions (Lu et al., 2011; Zheng et al., 2014), our regional calibration model has several features: (1) samples are only from lake surface sediments, rather than multiple sedimentary environments; (2) samples are distributed more evenly in space, rather than along the main roads; (3) RMSEP (root mean square error of prediction) for mean annual precipitation is lower. Therefore, our calibration model can work well in precipitation reconstruction for the central and western TP. However, it is a regional calibration model, and can be not used for broad areas like other large-scale calibration models.

Fossil pollen assemblages and concentrations from Lake Tangra Yumco revealed that the desert vegetation was replaced by steppe vegetation during the last deglaciation, and alpine steppe persisted in the basin during the Holocene (Ma et al., 2019, 2020). The results indicated that arid or semi-arid climatic condition persisted since 17.5 cal kyr BP. Therefore, the regional transfer function for $P_{ann}$ can be used for this record. Statistical significance tests showed that the $P_{ann}$ reconstruction for Lake Tangra Yumco explained more of the variance in the fossil data than 95% of the reconstructions from transfer functions trained on random data (Telford and Birks, 2011). The results indicate that the $P_{ann}$ reconstruction for Lake Tangra Yumco is statistically significant, and the reconstruction is reliable using the WA-PLS model (Figure 5).

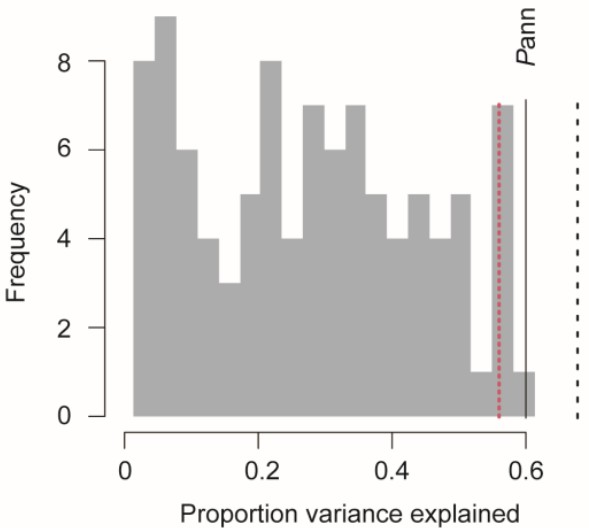

**Figure 5.** Statistical significance test of Pann reconstruction for Lake Tangra Yumco. Histogram showing the proportion of the variance in the pollen record of Lake Tangra Yumco explained by reconstructions of 99 $P_{ann}$ calibration functions trained with random data. The dotted red line represents the proportion of variance explained by random data-trained calibration functions at the 95% confidence level. The black solid line indicates the proportion of variance explained by the reconstructed $P_{ann}$. The dotted black line marks the proportion of variance explained by the first axis of the principal component analysis (PCA).

The application of the WA-PLS transfer function for $P_{ann}$ to the Lake Tangra Yumco pollen data is the first quantitative record in the central-western TP, located in the Indian summer monsoon margin. To ensure the effective number of species

occurrences and uniformity, predicted $P_{ann}$ values for fossil samples with Hill's N2 > 4 (Hill, 1973) were used in the reconstruction result (Figure 6). The pollen-derived precipitation record shows that the lake basin was most arid from 17.5 to 16.1 cal kyr BP with < 100 mm of $P_{ann}$. During 16.1-13.2 cal kyr BP, $P_{ann}$ increased rapidly up to > 200 mm. $P_{ann}$ showed a peak value (up to 300 mm) at approximately 13 cal kyr BP, followed by a low value of 240 mm at approximately 12 cal kyr BP. $P_{ann}$ was highest values in the early and middle Holocene (11.6-5 cal kyr BP), which is 70 mm (nearly 30%) higher than modern annual precipitation on average. After 5 cal kyr BP, $P_{ann}$ showed a decreasing trend.

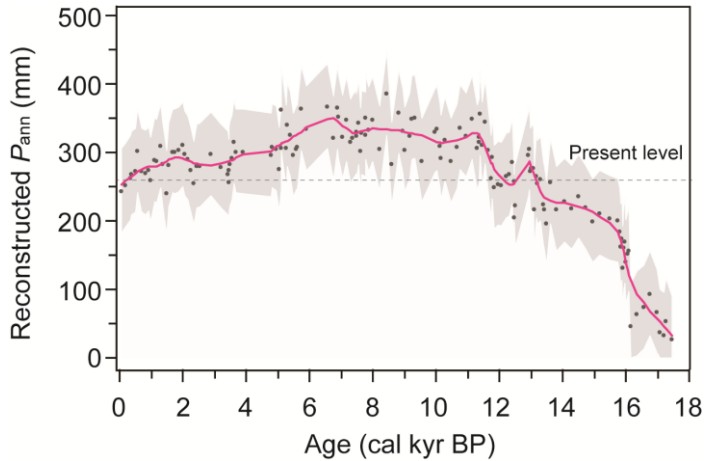

**Figure 6.** Predicted annual precipitation ($P_{ann}$) for Lake Tangra Yumco using the weighted averaging partial least squares (WA-PLS) inference model. The dotted line represents the modern annual precipitation of the basin. Grey shading indicates the ranges from predicted values subtracting the root mean square error of prediction (RMSEP) to predicted values adding the RMSEP.

## 6 Data availability

The dataset includes pollen counts and pollen percentages of the lake surface samples. Locations (latitude, longitude and altitude) and climatic data for each lake site are also given in the dataset. This dataset can be openly accessible via Zenodo portal: https://doi.org/10.5281/zenodo.8008474 (Ma et al., 2023).

## 7 Summary

We present and analyse a pollen dataset based on surface sediments from 90 lakes covering the central-western TP. This dataset includes pollen counts and percentages, as well as latitude, longitude, altitude and corresponding climate data. Ordination analyses indicate that mean annual precipitation ($P_{ann}$) is the dominant climatic parameter controlling variations in the modern pollen distribution of the dataset on the central-western TP. A quantitative transfer function was developed to estimate $P_{ann}$ from the modern pollen dataset using the Weighted Averaging Partial Least Squares (WA-PLS). Our results

indicate that the regional pollen dataset can provide useful and reliable quantitative estimates of past precipitation change in the central and western TP. The pollen-derived transfer function was tested for a pollen record on the central TP, which is the first quantitative paleo-precipitation record in the Indian summer monsoon margin.

Most of our lake sites are located in extremely remote areas with difficult access. The pollen data from these localities can fill the geographical gap left by other published modern pollen datasets and make the samples evenly distributed in the combined pollen dataset. Apart from its use for quantitative precipitation reconstructions in the central and western TP, our dataset can also be combined with other pollen datasets to improve the reliability of quantitative climate reconstructions across the entire TP.

**Author contributions**

LZ designed the study. QM, JJ, JW, YW, and LH collected the samples. QM made the pollen extraction and identification, and drafted the manuscript. JJ, JW, YW, LH and TH revised the manuscript.

**Competing interests**

The contact author has declared that none of the authors has any competing interests.

**Disclaimer**

Publisher's note: Copernicus Publications remains neutral with regard to jurisdictional claims in published maps and institutional affiliations.

**Acknowledgements**

We thank all colleagues and students in the field for their help with sampling. We would like to thank Prof. Qinghai Xu, Prof.
Lingyu Tang and Dr. Xinmiao Lü for their help with sample treatment and pollen identification. We also acknowledge Dr. Ruimin Yang for her assistance in producing diagrams. Climate data were provided by the National Tibetan Plateau Data Center.

**Financial support**

This research has been supported by the National Natural Science Foundation of China (41831177, 42272223), the Second
Tibetan Plateau Scientific Expedition and Research (STEP) (2019QZKK0202), and the Innovation Program for Young Scholars of TPESER (TPESER-QNCX2022ZD-01).

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
