# Peer review of "A modern pollen dataset from lake surface sediments on the central-western Tibetan Plateau"

_Earth System Science Data, 2023_

## Author Comment (AC1)

**Reply to Referee #1**

*The strength of the paper and the dataset is that the modern pollen surface samples are from lake sediments. This is an edge compared to most other modern pollen datasets from China and other regions. The procedure for using this dataset for constructing a pollen-climate calibration model is described in the paper, and it follows a standard routine with WA-PLS-based quantitative climate reconstructions. The data and the results are generally clearly presented, although the paper is very short with hardly any relevant discussion.*

*As a data description, the paper can be fine as such, but it would be interesting to develop the study in the future. The authors stress the importance of having both the modern pollen samples and fossil pollen samples from the same sedimentary environment, lakes in this case, and I agree with this. Consequently, it would be interesting to test whether the current calibration model works better as compared to the calibration models based on samples collected from varying sedimentary environments, such as topsoils, moss polsters etc.*

Response: We thank the reviewer for the good suggestion. Compared to pollen data from varying sedimentary environments (topsoils, moss polsters etc.), modern pollen data for the calibration model should be extracted preferably from lake surface sediments if possible (Birks et al., 2010). The central-western Tibetan Plateau contains a large number of lakes, providing an opportunity to develop a regional calibration model based on modern pollen samples from lake surface sediments. So far, there is no specific calibration model for the central and western Tibetan Plateau based on samples collected from different sedimentary environments. In fact, palynologists have successfully established pollen-climate transfer functions based on samples from varying sedimentary environments in the Tibetan Plateau and its surroundings(Lu et al., 2011), or even in China (Zheng et al., 2014), which includes our study area. These models can be used to make quantitative climate inferences based on pollen records from large areas. Compared with these large-scale calibration models, our pollen-climate transfer function has several features: (1) samples are only from lake surface sediments, rather than from multiple sedimentary environments; (2) samples on the central and western Tibetan Plateau are more evenly distributed in space, rather than along the main roads; (3) our RMSEP (root mean square error of prediction) for mean annual precipitation is lower. Consequently, our calibration model can work well in precipitation reconstruction for the central and western Tibetan Plateau. Of course, our dataset contains a narrower ecological and climatic gradient. Therefore, it can be considered as a regional calibration model for the central and western Tibetan Plateau, and can be not used for broad areas like other large-scale calibration models.

Considering the reviewer's comments, we added the related text in "5 Pollen-climate transfer function construction and application": "Our pollen-climate transfer function can be considered as a regional calibration model for the central and

western TP. So far, there is no specific calibration model for this region. Compared with the large-scale calibration models covering our study region and other regions (Lu et al., 2011; Zheng et al., 2014), our regional calibration model has several features: (1) samples are only from lake surface sediments, rather than multiple sedimentary environments; (2) samples are distributed more evenly in space, rather than along the main roads; (3) RMSEP (root mean square error of prediction) for mean annual precipitation is lower. Therefore, our calibration model can work well in precipitation reconstruction for the central and western Tibetan Plateau. However, it is a regional calibration model, and can be not used for broad areas like other large-scale calibration models."

*Another interesting angle for the study would be to apply some novel transfer function approach in the study. In Fig. 4 we can see that the calibration model has a quite serious bias towards too low values at the upper end of the precipitation gradient. The authors briefly comment on this underestimation of high values. Such an edge effect is a typical problem with WA-PLS-based transfer functions. An amendment has been suggested with the use of tolerance-weighted WA-PLS (Liu et al., 2020) and it would be really interesting to see whether the use of tolerance-weighted WA-PLS would improve the edge effect problem in the calibration model and influence the precipitation reconstruction shown in Fig. 6.*

Response: Thanks for the highly specialized comments. Liu et al. (2020) motivated an improved version of WA-PLS by using the information on the climatic tolerances of taxa, and further considering the frequency (*fx*) of the climate variable in the training dataset. Due to the relatively small sample size in our study, we just took into account the climatic tolerances of pollen taxa, without utilizing the *fx* correction. That's because *the fx*-corrected criteria set by Liu et al. (2020) would remove many geographically and climatically close sites in our training dataset, which would lead to an excessively small training set for reconstruction.

Model performance of WA-PLS and TWA-PLS without *fx* correction for mean annual precipitation is shown in Table 4. All modern pollen samples are used in these methods. The last significant number of component for WA-PLS or TWA-PLS is considered to be the appropriate component for quantitative reconstruction (Liu et al., 2020). Therefore, the performances of WA-PLS component 2 and TWA-PLS component 2 were compared. TWA-PLS has RMSEP of 62.262, while WA-PLS has RMSEP of 61.381. TWA-PLS has $R^2$ of 0.675, while WA-PLS has $R^2$ of 0.684. In general, compared with WA-PLS, tolerance-weighted WA-PLS without *fx* correction did not improve model performance based on our pollen dataset. One explanation could be that our sample size is not large enough and the sample sites are mainly distributed in the middle and lower parts of the precipitation gradient with a relatively uniform distribution. Finally, we still use the WA-PLS method to develop the pollen-precipitation transfer function for our dataset.

The related text was added in "3.3 Numerical analyses" and "5 Pollen-climate

transfer function construction and application".

In 3.3: "weighted averaging partial least-squares regression (WA-PLS) and tolerance-weighted WA-PLS (TWA-PLS) (Liu et al., 2020). TWA-PLS is an improved version of WA-PLS by considering the information about the climatic tolerances of taxa (Liu et al., 2020). TWA-PLS was performed using the function TWAPLS.w function in the package fxTWAPLS version 0.1.2 (Liu et al., 2020) for R 4.2.3."

In 5: "Performances of various calibration models are shown in Table 4. The last significant number of component for WA-PLS or TWA-PLS is considered to be the suitable component for each method. Thus, WA-PLS component 2 ($p$ = 0.031) and TWA-PLS component 2 ($p$ = 0.047) are used to compare with other methods. WA-PLS component 2 and TWA-PLS component 2 show better performances than WA, MAT and WMAT, due to their lower RMSEP and higher $R^2$. Further comparison between WA-PLS and TWA-PLS reveals that WA-PLS component 2 has a lower RMSEP and a higher $R^2$. It means that TWA-PLS does not improve model performance based on our pollen dataset, compared with WA-PLS. One explanation could be that our sample size is not large enough and sample sites are mainly distributed in middle and lower parts of the precipitation gradient with a relatively uniform distribution. Finally, we still use WA-PLS method to develop the pollen-precipitation transfer function for our dataset."

*Some minor remarks*

*(1) page 2 line 36 "reconstruction of climate data" remove "data"*

Response: We have deleted it.

*(2) page 2 line 51 remove "desperately"*

Response: We removed it.

*(3) page 3 as this is a dataset paper, it would be better to include key data from all 90 sites, such as location, altitude, climate etc.*

Response: In fact, the information including location, altitude and climate data of 90 lake sites is available in the Zenodo portal: https://doi.org/10.5281/zenodo.8008474 (Ma et al., 2023), accompanied by pollen data of this study. Considering the reviewer's comments, we added the related text in the Abstract: "Data from this study, including pollen data for each sample and information on the sampled sites (location, altitude and climate data), are openly available via the Zenodo portal (Ma et al., 2023; https://doi.org/10.5281/zenodo.8008474)".

In addition, we added a sentence in "2 Study area": "A total of 90 lakes were sampled in the study region, which information is presented in Ma et al. (2023)".

*(4) page 4 Fig. 1 add an index map*

Response: According to the reviewer's comments, we modified Fig. 1.

[Figure]

**Figure 1**. Study area and sampling sites of modern lake sediments. (a) Location of the study area; (b) spatial distribution of sampling lake sites (black dots) and annual precipitation; (c) vegetation map of the Tibetan Plateau with the location of Lake Tangra Yumco (red dot).

*(5) page 5 lines 103-104. This in unclear. What does "reanalysis datasets" mean? And how were the altitudinal differences handled? Was the windward or leeward side location of the lakes in relation to the mountains considered?*

Response: The climate data used in our study are obtained from the Chinese Meteorological Forcing Dataset, which has been described in a data descriptor paper (He et al., 2020). Considering the reviewer's comments, we added the relevant text to clarify "reanalysis datasets": "The gridded reanalysis/remote sensing data are GLDAS NOAH10SUBP 3H, GLDAS NOAH025 3H, Modern Era Retrospective-Analysis for Research and Applications (MERRA) MAI3CPASM 5.2.0 20, Global Energy and Water Exchanges – Surface Radiation Budget (GEWEX-SRB) REL3.0 SW 3HRLY 21, and TRMM 3B42 v7".

We also added the text "The dataset introduced high-resolution elevation data into the interpolating process of particular variables, such as air temperature". The authors first calculated the sea-level temperature for the observational and reanalysis data, respectively, and then merged the two types of data. Finally, they calculated the

air temperature at the altitude of the land surface using high-resolution terrain elevation data. We extracted the climate data from the Chinese Meteorological Forcing Dataset. The dataset has a spatial resolution of 0.1, and we did not further consider the windward or leeward side location of the lakes.

*(6) page 6 add citations to the selection of RDA as the linear method and add citations to the use of VIF to check for collinearity.*

Response: Related references were added in the text.

The references are:

ter Braak, C.J.F., Prentice, I.C.: A theory of gradient analysis. Adv. Ecol. Res., 18, 271–317, https://doi.org/10.1016/S0065-2504(08)60183-X, 1988.

ter Braak, C.J.F., Verdonschot, P.F.M.: Canonical correspondence analysis and related multivariate methods in aquatic ecology. Aquat. Sci., 57, 255–289, https://doi.org/10.1007/BF00877430, 1995.

ter Braak, C.J.F.: Canoco–a FORTRAN program for canonical community ordination by (partial) (detrended) (canonical) correspondence analysis, principal components analysis and redundancy analysis (version 2.1). Technical Rep.LWA-88-02. GLW, Wageningen, 95 pp, 1988.

ter Braak, C.J.F., and Šmilauer, P.: CANOCO reference manual and user's guide: software for ordination (version 5). Microcomputer Power Ithaca, 2012.

*(7) page 7 line 137 "a novel method" delete "novel". What was novel in 2011 is not novel any more.*

Response: We deleted the word.

*(8) page 9 It would have been better not to remove this one outlier site from the dataset. While outliers are sometimes deleted this way, it is a questionable thing to do. Firstly, it is an easy trick to improve performance statistics by removing the "dodgy" samples. Secondly, the performance statistics of the current dataset (e.g. $R^2$ values) cannot be compared directly with other datasets in which no samples have been removed.*

Response: According to the reviewer's comments, we have re-developed the pollen-precipitation transfer function by using all samples. The related text, table and figures were also modified.

**References:**

Birks, H. J. B., Heiri, O., Seppä, H., Bjune, A. E.: Strengths and weaknesses of quantitative climate reconstructions based on late-Quaternary biological proxies, Open Ecol. J., 3, 68–110, https://doi.org/10.2174/1874213001003020068, 2010.

He, J., Yang, K., Tang, W., Lu, H., Qin, J., Chen, Y., and Li, X.: The first high-resolution meteorological forcing dataset for land process studies over China, Sci. Data, 7, 25, https://doi.org/10.1038/s41597-020-0369-y, 2020.

Liu, M., Prentice, I. C., ter Braak, C.J.F., and Harrison, S.P.: An improved statistical approach for reconstructing past climates from biotic assemblages, Proc. R. Soc. A, 476, 20200346, https://doi.org/10.1098/rspa.2020.0346, 2020.

Lu, H., Wu, N., Liu, K., Zhu, L., Yang, X., Yao, T., Wang, L., Li, Q., Liu, X., Shen, C., Li, X., Tong, G., and Jiang, H.: Modern pollen distributions in Qinghai-Tibetan Plateau and the development of transfer functions for reconstructing Holocene environmental changes, Quat. Sci. Rev., 30, 947–966, https://doi.org/10.1016/j.quascirev.2011.01.008, 2011.

Ma, Q., Zhu, L., Ju, J., and Wang, J.: A modern pollen dataset from lake surface sediments on the central-western Tibetan Plateau [Data set], Zenodo, https://doi.org/10.5281/zenodo.8008474, 2023.

Zheng, Z., Wei, J., Huang, K., Xu, Q., Lu, H., Tarasov, P., Luo, C., Beaudouin, C., Deng, Y., Pan, A., Zheng, Y., Luo, Y., Nakagawa, T., Li, C., Yang, S., Peng, H., and Cheddadi, R.: East Asian pollen database: modern pollen distribution and its quantitative relationship with vegetation and climate, J. Biogeogr., 41, 1819–1832, https://doi.org/10.1111/jbi.12361, 2014.

---

## Author Comment (AC2)

**Reply to Referee #2**

*In this manuscript, the authors present a modern pollen dataset based on surface sediments from 90 lakes covering the central and western Tibetan Plateau, and applied it to reconstruct the paleoprecipitation of a fossil pollen record from Lake Tangra Yumco on the central TP. This study provides a significant dataset for modern pollen research in the TP. The results of the investigation are important and this is a very interesting paper, I believe upon update will be a good article. Therefore, I suggest that some minor issues to be improved.*

*1) In the Study area, only the climate and vegetation are included, adding the geological and geomorphological information would provide a more comprehensive understanding, because these may also affect pollen assemblages.*

Response: Thanks for the comments. We have added the related text in "2.1 Topography and climate": "The TP is the broadest and highest elevation collisional system of the Earth (Ding et al., 2022). Under the long-term comprehensive effect of the internal and external agency, the TP manifests extremely complex topographic features. Our study region is located in the central and western TP between 28.5°- 35.2°N and 79.8°- 91.5°E. Several nearly west-east trending mountain ranges are distributed in the region, which are the Himalayas, the Gangdese-Nyainqentanglha Mountains, the Karakorum-Tanggula Mountains and the Kunlun Mountains from south to north. Between these mountain ranges there are several broad valleys, which are divided by the secondary tectogenesis into many sedimentary basins of different sizes. A large number of lakes lie on the bottom of the basins (ITCAS, 1988)".

*2) Quercus E and Quercus D should be marked with their full names when they first appear in the text.*

Response: According to the reviewer's comments, we corrected them.

*3) The sum of Variance explained of Pann, TJan and TJuly in RDA results is only 35.3%. Is there any influence of other factors considered?*

Response: We agree that there may be other climatic or environmental factors influencing the modern pollen distribution. We select these variables because they are considered as important climatic factors for plant distribution and growth, and have been widely used for palaeoclimate reconstructions based on pollen data (Shen et al., 2006; Herzschuh et al, 2010; Cao et al., 2021). According to previous quantitative pollen-climate studies over the Tibetan Plateau, these factors, or some of them, are the only factors considered in most of the studies (Shen et al., 2006; Herzschuh et al., 2010; Wang et al., 2014; Cao et al., 2021). To increase the comparability of our analysis process with other studies, we also selected these factors as target variables. Partial RDA shows that our reconstructed target variable-$P_{ann,}$ as a sole variable,

explains 24.2% of the variation in the pollen data, which also reveals the importance of $P_{ann}$ for modern pollen distribution in our study region.

*4) Line 212, What the "Hill"s N2" means?*

Response: Hill's N2 diversity (Hill, 1973) was used to measure the effective number of pollen taxa occurrences (Juggins, 2007). Fossil pollen samples with low Hill's N2, such as several samples older than 16.2 cal kyr BP in our records, could lead to inaccuracy of climate reconstructions. Therefore, we set the threshold to eliminate the reconstruction results of the fossil samples with the relatively rare pollen taxa.

*5) "This dataset can be openly accessible via Zenodo portal: https://doi.org/10.5281/zenodo.8008474", the URL cannot be opened, please check whether the URL is correct.*

Response: We have checked the URL by using different browsers (Microsoft Edge, Internet Explorer and Google), and found that it is correct. Perhaps the Zenodo website was under maintenance at the time you visited, or for some other reason. Please try again.

**References:**

Cao, X., Tian, F., Li, K., Ni, J., Yu, X., Liu, L., and Wang, N.: Lake surface sediment pollen dataset for the alpine meadow vegetation type from the eastern Tibetan Plateau and its potential in past climate reconstructions, Earth Syst. Sci. Data, 13, 3525–3537, https://doi.org/10.5194/essd-13-3525-2021, 2021.

Ding, L., Kapp, P., Cai, F., Garzione, C.N., Xiong, Z., Wang, H., and Wang, C.: Timing and mechanisms of Tibetan Plateau uplift. Nat. Rev. Earth Env., 3, 652–667, https://doi.org/10.1038/s43017-022-00318-4, 2022.

Herzschuh, U., Birks, H. J. B., Mischke, S., Zhang, C., and Böhner, J.: A modern pollen-climate calibration set based on lake sediments from the Tibetan Plateau and its application to a late Quaternary pollen record from the Qilian Mountains, J. Biogeogr., 37, 752–766, https://doi.org/10.1111/j.1365-2699.2009.02245.x, 2010.

Hill M.O.: Diversity and evenness: A unifying notation and its consequences. Ecology, 54, 427–432, https://doi.org/10.2307/1934352, 1973.

ITCAS (Investigation Team of Chinese Academy of Science): Vegetation in Tibetan Plateau, Science Press, Beijing, 1988.

Juggins, S.: C2, Software for ecological and palaeoecological data analysis and visualisation, User guide Version 1.5. School of Geography. Politics & Sociology, Newcastle University, Newcastle, 2007.

Shen, C., Liu, K., Tang, L., and Overpeck, J. T.: Quantitative relationships between modern pollen rain and climate in the Tibetan Plateau, Rev. Palaeobot. Palynol., 140, 61–77, https://doi.org/10.1016/j.revpalbo.2006.03.001, 2006.

Wang, Y., Herzschuh, U., Shumilovskikh, L.S., Mischke, S., Birks, H.J.B., Wischnewski, J., Böhner, J., Schlütz, F., Lehmkuhl, F., Diekmann, B., Wünnemann, B., Zhang, C.: Quantitative reconstruction of precipitation changes on the NE Tibetan Plateau since the Last Glacial Maximum – extending the concept of pollen source area to pollen-based climate reconstructions from large lakes, Clim. Past, 10, 21–39, https://doi.org/10.5194/cp-10-21-2014, 2014.